# Anti-Influenza Virus Activity of *Citrullus lanatus* var. *citroides* as a Functional Food: A Review

**DOI:** 10.3390/foods12203866

**Published:** 2023-10-22

**Authors:** Ryosuke Morimoto, Yuji Isegawa

**Affiliations:** 1Department of Health and Nutrition, Faculty of Human Life Science, Shikoku University, Tokushima 771-1192, Japan; rmorimoto@shikoku-u.ac.jp; 2Department of Applied Biological Chemistry, Graduate School of Agriculture, Osaka Metropolitan University, Sakai, Osaka 599-8531, Japan

**Keywords:** wild watermelon, phytochemical, phytoestrogen, flavonoid, influenza virus

## Abstract

Influenza is an acute respiratory illness caused by the influenza virus, in response to which vaccines and antiviral drugs are administered. In recent years, the antiviral effects of plants and foods have garnered attention. This review is the first to summarize the therapeutic properties of wild watermelon (*Citrullus lanatus* var. *citroides)* against influenza from a phytochemical viewpoint. Wild watermelon is a wild plant with significant potential as a therapeutic candidate in antiviral strategies, when focused on its multiple anti-influenza functionalities. Wild watermelon juice inhibits viral growth, entry, and replication. Hence, we highlight the possibility of utilizing wild watermelon for the prevention and treatment of influenza with stronger antiviral activity. Phytochemicals and phytoestrogen (polyphenol, flavonoids, and prenylated compounds) in wild watermelon juice contribute to this activity and inhibit various stages of viral replication, depending on the molecular structure. Wild plants and foods closely related to the original species contain many natural compounds such as phytochemicals, and exhibit various viral growth inhibitory effects. These natural products provide useful information for future antiviral strategies.

## 1. Introduction

Influenza is an acute respiratory disease caused by the influenza virus, which is a single-stranded negative-sense RNA virus, belonging to the family of *Orthomyxoviridae*. Its surface structure is composed of a spike protein, such as hemagglutinin (HA), and neuraminidase (NA). Influenza particles contain an envelope of lipid membranes, which directly covers a matrix protein (M protein). The influenza virus particle consists of eight viral genome segments that encode virus-specific proteins. These viral proteins and genes are key to the spread and transmission of influenza. Influenza viruses are classified into types A, B, and C depending on the antigenicity of the M protein and nucleoprotein (NP). Additionally, type D, which infects cattle and pigs, has been newly reported. Types A and B, in particular, are known to cause pandemics worldwide [1,2].

Type A is classified into multiple subtypes and strains based on the antigenic differences in the spike proteins, HA and NA; to date, 18 and 11 types have been reported, respectively, consisting of subtypes from H1N1 to H18N11. Type B is roughly divided into B/Victoria/2/87 and B/Yamagata/16/88, based on the antigenicity of HA [3].

After infection, symptoms such as fever, headache, general malaise, muscle pain, arthralgia, cough, and nasal discharge continue and begin to improve. Figure 1 illustrates the growth steps of influenza virus within mammalian cells. Initially, Influenza viruses recognize glycoproteins on the host cell membrane (Step 1). Virus particles are passively absorbed into the cell via endocytosis, the mechanism of foreign body uptake of the host cell (Step 2). Next, virus particles are cleaved of HA *via* acidification in intracellular endosomes (Step 3), following the membrane via uncoating (Step 4). The viral genome released into the cytoplasm moves into the nucleus as a ribonucleoprotein complex (RNP; Step 5), where virus-derived genomes are transcribed and replicated (Step 6). Virus-derived mature mRNA binds to the ribosome of the host cell and NP, RNA polymerase (PA, PB1, PB2), HA, NA, and the M protein are synthesized (Step 7). The synthesized RNA polymerase and NP are transported into the host nucleus to form viral RNPs, which are transported into the cytoplasm (step 8); they reassemble with membrane proteins and other compounds to form viral particles (step 9). Finally, new progeny viruses are released outside the cell (Step 10). Anti-influenza drugs have been developed that target each step in the viral replication process (Step 4—amantadine, Step 6—favipiravir and baloxavir, and Step 10—oseltamivir) for treatment against the influenza virus. 

Influenza pandemics are caused by mutations in the viral genome. Influenza A virus subtypes are mainly observed in avian reservoirs, including waterfowl species. These virus subtypes can be transmitted to birds, poultry, pigs, and a variety of other mammals. Occasionally, cases of avian influenza virus infecting humans have also been reported [4]. Therefore, influenza caused by influenza virus infection is a potential zoonotic disease. The type A(H1N1) pdm09 subtype that caused the pandemic in 2009 was reported as a new triple reassortant virus containing human, avian, and swine viral gene segments [5]. This virus spread throughout the world as a seasonal influenza, causing much damage to humanity as an infectious disease. The 1918 pandemic remains the worst infectious disease outbreak in history [6]. As COVID-19 continues to spread worldwide, “influenza” has been erased from the media; however, deaths from influenza continue to be reported [7,8]. Influenza viruses constantly undergo antigenic mutations and change their transmissibility and drug susceptibility. Although this has happened before, the concern is that it will happen again. As mentioned in the viral growth steps, several antiviral drugs have been developed, including oseltamivir, zanamivir, peramivir, laninamivir octanoate hydrate, and baloxavir, which are used globally. However, even if antiviral drugs are developed and approved, resistant viruses will still be reported within several years to decades. Most influenza virus strains have been reported to be drug-resistant to the antiviral drug amantadine [9]. There have been reports of influenza strains resistant to oseltamivir (NA inhibitor), which is used worldwide [10]. Oseltamivir, which is currently the antiviral drug most commonly used against influenza, may be restricted to some people [11,12]. Baloxavir differs from neuraminidase inhibitors (oseltamivir and others) that release the virus from the infected cell surface. Baloxavir inhibits intracellular cap-dependent endonucleases, and is attracting attention as a new drug that inhibits the viral mRNA replication step. Unfortunately, there are reports of the emergence of PA/I38X-substituted viruses and decreased susceptibility after treatment with baloxavir [13,14,15]. Similarly, viral strains resistant to previously developed antiviral drugs have been reported [16,17,18], which are caused by viral antigenic mutations. Current therapies may not be quickly responsive to the emergence of resistant viral strains. Therefore, the development of novel alternative therapies should also be considered. In recent years, studies on combinations of anti-influenza virus drugs and natural ingredients have been reported [19,20]. To adapt to viral antigenic mutations, we propose using multiple natural ingredients in antiviral strategies. As RNA virus infections garner growing attention worldwide, research on antiviral effects derived from natural products is of high importance, and research on food extracts and ingredients will continue to be reported. We focus on non-nutrient ingredients of plants as a source of naturally derived components for future antiviral strategies. 

The term phytochemical is derived from the Greek word “phyto”, meaning plant, and is defined as a non-nutritive, physiologically active substance contained in plants such as vegetables and fruits [21,22]. Phytochemicals are classified into several categories (Figure 2) and exhibit various functionalities. Herein, we will introduce phytochemicals that have physiological effects on humans, based on previously published work. Nakamoto et al. reported the relationship between the intake of soybeans, soybean products, and isoflavones, which are polyphenols [23]. This study assessed the intake of beans, soy products, and soy isoflavones using a 3-day dietary diary, and showed that intake of soybeans and soy isoflavones could reduce the risk of cognitive impairment in elderly Japanese women. The functionality of flavonoid-isoflavones has been well studied, and they are one of the more interesting compounds, with not only estrogenic effects but also non-estrogenic effects extensively reviewed by Alshehri et al. [24]. Sulfur-containing compounds derived from plants have also been reported. Wu et al. reported that *Moringa oleifera*, a member of the Moringa family, contained 4-(α-L-rhamnosyloxy) benzyl isothiocyanate, also known as Moringa isothiocyanate. Moringa isothiocyanate has been reported to inhibit cancer growth and promote cancer cell apoptosis through multiple signaling pathways, and has also been shown to have anti-chronic-disease effects [25]. Yao et al. found that terpenoids regulated lipid metabolism disorders, insulin resistance, and oxidative stress *via* regulating the AMPK, PPAR, and Nrf-2 pathways, and were effective in the treatment of non-alcoholic fatty liver disease [26]. 

Plants and their ingredients have a wide range of functionalities, and phytochemicals regulate biological functions through signal transduction. Researching the effects of phytochemicals on host transcription factors is expected to be useful in a wide range of fields, including research on antiviral strategies.

Food extracts such as cocoa, onion, and berries have also been reported to have effective anti-influenza virus effects [27,28,29]. Further, we reported on the functions of foods and their components in relation to antiviral activity. We have also reported the function of *Euglena* as a functional food [30], whereas this study reports a potential method to rapidly and efficiently improve the antiviral effect of Euglena. Previously, we showed that an antiviral component of soybean is daidzein [31] and revealed a part of its mechanism of action and signal transduction [32,33]. The antiviral activity and major components of Sacna (*Seriaceae*) [34], along with its strong antiviral activity, and the inhibition mechanism of wild watermelon (*Citrullus lanatus* var. *citroides*) have also been investigated [35,36]. Other studies revealed the relationship between different flavonoid structures and their antiviral activities [37], and the antiviral effect *via* flavonoid molecular modification [38]. These food extract effects are dependent on phytochemicals, their quantities, and their types. Food extracts consist of many active ingredients, of which a class of polyphenols is included. Several ingredients have been reported in terms of their inhibition of virus propagation. Flavonoids and apigenin, which is polyphenol, inhibited the replication process of the influenza virus [39]. The study reported that apigenin significantly suppressed the production of pro-inflammatory cytokines and interferons (IFN-*β* and IFN-*λ*1) and NA activity involved in viral budding. Luteolin, a structural analog of apigenin, was also found to suppress the expression of protein complexes associated with viral endocytic pathways [40]. The antiviral effects of natural constituents are diverse. Numerous types of flavonoids are expected to be useful in antiviral strategies. Based on previous research results, these constituents may be a useful approach for future antiviral strategies. The enrichment of these studies will enable their broad application to current antiviral strategies. Thus, this review focuses on specific plants as potential alternative therapies to existing viral drugs. We focused on the potent anti-influenza virus activity of wild watermelon (*Citrullus lanatus* var. *citroides*). Phytochemicals of wild watermelon are described and the future potential of wild watermelon as a functional food is advocated in this review. 

## 2. *Cucurbitaceae* Family and Wild Watermelon

Currently, members of the *Cucurbitaceae* are widely consumed. Common members of the cucurbit family include cucumbers, bitter melons, and watermelons. The peels and fruits of *Cucurbitaceae* foods, with the exception of watermelon, are bitter and bear an unpleasant odor. This bitterness and grassy smell may be involved in the functionality of foods [41,42]. As shown in the supplemental data [35], our *in vitro* studies have indicated that some cucurbit extracts have antiviral effects. Hot water extract of pumpkin (*Cucurbita moschhata*) and zucchini (*Cucurbita pero* L.) had IC_50_ values of 0.74 mg/mL and 1.07 mg/mL, respectively. 

Watermelons are spherical or oval in shape and come in a wide range of varieties, including those with green and black stripes, varieties without stripes, and varieties with a deep green rind color. The watermelon fruit is classified as a fruit vegetable in nutritional terms, and most of the varieties currently cultivated and distributed have sweet fruit characteristics. Watermelon fruit is a promising source of the amino acid L-citrulline [43]. It has been reported that the intake of citrulline through the ingestion of watermelon promotes the synthesis of nitric oxide, which has a vasodilatory effect [44]. As described above, watermelon fruit has been reported to have beneficial components with effects on human health, but our *in vitro* studies did not confirm any anti-influenza virus activity in watermelon juice. On the other hand, wild watermelon showed high antiviral activity.

Wild watermelon is a plant native to the Kalahari Desert in southern Africa; it grows in harsh environments such as deserts, and retains moisture in the fruit, thus obtaining the name, “desert water jar,” from local nomads. Although the fruit of the commercial watermelon species is red, the fruits can be white, yellow, and green, and are characterized by their low sugar content, viscosity, and caloric content. Commercially, studies have reported improvements in fruit juice quality [45]. Reports on the functionality (anti-inflammatory properties) of wild watermelon emerged in the 2000s. The high antioxidant capacity of wild watermelon is due to the accumulation of antioxidants, proteins, and amino acids, such as metallothionein and citrulline [43,46,47]. In addition, its effects on humans have been reported, such as the influence of its acute ingestion on atherosclerosis by Fujie et al. [48]. Although the high antioxidant properties, components, and effects of wild watermelon on atherosclerosis have been reported, other functional components and effects remain unknown. Additionally, no reports of its efficacy against the influenza virus have been made. Wild watermelon represents a relatively new research target in antiviral therapy, and its ingredients were first identified by our research group.

## 3. Anti-Influenza Virus Activity of Wild Watermelon Juice (WWMJ) and Its Flavonoids

We reported the anti-influenza virus activity of WWMJ *in vitro* and *in vivo* [35]. Our previous study reports the novel functionality of WWMJ from the perspective of antiviral effects, evaluating its effects on MDCK cells, a common cell line in influenza virus research, and infected mice. Additionally, an *in vitro* evaluation of WWMJ and its flavonoids showed that compared to *Cucurbitaceous* food extracts, it had inhibitory effects on all virus strains at a low concentration of 1 mg/mL or less. That is, the effective concentration in cultured cells was 0.11 to 0.20 mg/mL for the A(H1N1) strain. Interestingly, WWMJ inhibited the viral growth of two oseltamivir-resistant strains of the influenza virus (Osaka/2024/2009, Osaka/71/2011 strains). The IC_50_ concentrations were 0.13 and 0.20 mg/mL, respectively. In addition, it was effective against type A (H3N2) and type B (Yamagata and Victoria strains). The effective concentrations were 0.41 to 0.88 mg/mL. No toxicity was observed in MDCK cells, even at a concentration 10 times the IC_50_ value. WWMJ may be more flexible in coping with viral mutation than antiviral drugs that specifically inhibit viral replication steps. Therefore, we considered its application in humans in the future. We examined the efficacy of WWMJ against infected mice using BALB/c mice. *In vivo* experiments confirmed that the nasal administration of WWMJ improved the survival rate of virus-infected mice compared to the phosphate-buffered saline control group. WWMJ treatment increased the lifespans of influenza virus-infected mice by at least 2–3 days. It also suppressed inflammation in the lungs of infected mice, suggesting that the functional components of WWMJ suppress the inflammation of lung cells caused by influenza virus infection. *In vitro* and *in vivo* results support the potential of this useful plant in antiviral strategies. 

The time-of-addition assay indicated the WWMJ inhibition mechanism. WWMJ inhibits the influenza virus growth steps, entry (−1–0 h, adsorption), and replication (4–8 h, late stage). Because “adsorption” in this experimental system includes the process of the viral recognition of sugar chains on the cell membrane and endocytosis, additional detailed experiments were required. After investigation, the experimental results showed that rather than inhibiting the viral recognition of sugar chains on the cell membrane and HA protein membrane fusion, WWMJ was associated with energy-dependent endocytosis. The viral particle uptake mechanism (virus replication step 2) is a biologically significant process. Influenza viruses utilize the physiological mechanism of the host cell, which is energy-dependent endocytosis, to enter cells. Viral uptake processes are known in micro pinocytosis, clathrin-mediated endocytosis, and caveolin-mediated endocytosis, which also involves influenza virus entry [49]. WWMJ constituents may be involved in viral entry [35], so investigating the targeting genes and proteins involved in endocytosis is necessary. 

The period of 4–8 h after viral infection in mammalian cells includes viral transcription and replication, protein synthesis, assembly, and particle release processes. WWMJ may inhibit one or more of the viral growth processes. WWMJ constituents are expected to have multiple anti-influenza virus effects. Several studies have reported that plant phytochemicals could function as activity inhibitors of influenza virus spike proteins (HA and NA) and inhibit the expression of viral genes and other protein syntheses [50,51,52]. Therefore, we presumed that the modification groups of flavonoid molecules could impact the anti-influenza virus effect. We determined that flavonoids were important in viral strategy and investigated the antiviral effects of various flavonoids [37]. To investigate the effect of modifying groups in the flavonoids, we focused on the hydroxyl group and methoxy group and prepared 22 standard flavonoids. Among them, 19 flavonoids showed antiviral activity. The flavonoids that lost activity were genistein (isoflavone), and pinocembrin (flavanone). The structures of these compounds suggested the importance of the position of the hydroxyl group modifying the C and B-rings. In the 19 flavonoids that showed antiviral activity, factors such as the number and position of hydroxyl groups were closely related. For example, many of the flavonoid class compounds examined exhibited antiviral activity when the B ring was modified with a hydroxy group or a methoxy group, confirming the importance of the B ring in terms of molecular structure. Our results revealed that the B-ring in the flavonoid molecule had a catechol structure; hence, from the structure of the hydroxyl group and methoxy group, the electron-donating property in the flavonoid molecule is also expected to be related to the viral protein or other protein interactions. Also, we reported that the anti-viral activity of flavonoids and the flavonoid skeleton commonly inhibited the late stage of influenza virus replication, and flavonols and flavanones could inhibit the virus entry, showed growth inhibitory effects on oseltamivir-resistant strains. Different structures of the flavonoid skeleton had different inhibitory effects on the influenza virus growth. WWMJ significantly inhibited the late stage of viral growth (4–8 h). As a plant, WWMJ may also contain various types of phytochemicals that are expected to contribute to the antiviral effect. 

## 4. Constituents of WWMJ

Plants contain numerous phytochemicals whose composition and content vary depending on the species. Flavonoids are categorized into subclasses depending on the basic structure. Soybeans contain flavonoids termed isoflavones [53] and onions contain quercetin [54]. The natural compounds in plant species have been characterized. As described in Section 2, wild watermelon is the original watermelon species, differing in properties from the common watermelon. This section describes the difference between wild watermelon and watermelon in terms of sugar content and phytochemical composition, including flavonoids. First, we evaluated the quantity of sugar contained in wild watermelon and watermelon using Brix concentration. The sugar contents of wild watermelon and watermelon were found to be 2°Bx and 8°Bx, respectively. The sugar quantity contained in watermelon was four times that in wild watermelon. Next, to investigate the constituents of wild watermelon in detail, we performed a metabolome analysis [36]. From the *m*/*z* value in Figure 3, many phytochemicals were confirmed to be low-molecular-weight constituents rather than high-molecular-weight constituents. The vertical axis shows the *m*/*z* value, that is, the dimensionless mass. The horizontal axis shows the mass detection time during analysis. Based on the analysis conditions, it was expected that the earlier the detection time, the more water-soluble constituents would be detected, and the later the detection time, the more fat-soluble constituents would be detected. Figure 3 shows that many constituents retained in the ODS column are eluted and detected with low-concentration organic solvents (acetonitrile). The analysis suggested that WWMJ may contain many water-soluble constituents. In an exhaustive analysis, flavones, flavonols, and flavanones were found to be abundant among the polyphenols, accounting for 31%, 16%, and 17% of the total polyphenols, respectively [36]. We detected some WWMJ constituents using a mass spectrometer (triple quadrupole, QQQ, Agilent technologies, Santa Clara, CA, USA) which were glycosylated flavonoids and aglycons. The detected polyphenol aglycon part and viral activity are shown in Table 1. From these results and reports [36], plant or WWMJ flavonoids are suggested to contribute to the anti-influenza virus activity. However, it is also necessary to consider the amount of these phytochemicals, the IC_50_ value, and the synergistic effect between phytochemicals. 

Originally, flavonoids were “foreign substances” for humans and viruses, and the immune response works appropriately. Flavonoids are also involved in the host immune response systems, such as the aryl hydrocarbon receptor and keap1-Nrf2 systems. In addition, the immune response and *in vivo* oxidation are closely related, and cellular oxidative stress is induced by daidzein [32]. Oxygen radicals such as superoxide, hydroxyl radicals, and lipid peroxyl radicals are known to increase during virus infection [55], and flavonoids can capture and eliminate them. Other studies have reported flavonoids as agents that protect the lungs from the adverse effects of oxygen-derived free radicals released during influenza virus infection [56], carriers of metal ions as ionophores [57], and 5-lipoxygenase [32,33], which are involved in lipid oxidation reactions. The Keap1-Nrf2 system plays an important function in the control mechanism of this biological response. Nrf2 is a transcription factor that is activated by oxidative stress such as reactive oxygen species and electrophilic substances, and is known to protect cells in various aspects. Therefore, we believe that this electrophilicity in the molecular structure of flavonoids is also necessary information for antiviral strategies. In this research field, there are numerous reports on the relationship between flavonoids and influenza viruses. There are reports that flavonoids enhance host cell immunity by activating transcription factors [58], inhibiting the activity of viral protein [59] and fusion of virus and cell membranes [60]. When considering the activities of flavonoids in detail, the electron density and tertiary structure of their molecular structures are important key properties in terms of antiviral strategies. 

## 5. Potential of Prenylated Flavonoids in Antiviral Strategies 

Specific plants contain prenylated compounds such as xanthohumol and 8-PN, and their functionality has attracted attention in medicine and physiology [61,62]. Seliger et al. reported that prenylated compounds are potent inhibitors of the aldo-keto reductase (AKR) superfamily of protein enzymes, which are pharmacological targets for cancer therapy and diabetic complications [63]. Shahinozzaman reported that the prenylated compounds contained in propolis have anti-inflammatory, anti-diabetic, and anti-Alzheimer effects [64]. The details were the inhibitory activity of enzyme proteins that are responsible for inducing each disease. Although various pharmacological effects have been reported [65,66,67], reports on the functionality of prenylated compounds have increased since the early 2000s. Therefore, reports on the anti-influenza virus effects of prenylated compounds and their derivatives are novel research. Thus, we focused on prenylated flavonoids in plants.

We identified 8-prenylnaringenin (8-PN) as a characteristic constituent of WWMJ. Its activity was confirmed to be approximately 10 times higher than that of the original compound, naringenin. This result suggests that flavonoid prenylation has a significant impact on antiviral efficacy. We considered the prenylation of flavonoids to be one of the important tools in antiviral strategies. To investigate the viral effect of prenylated flavonoids, we synthesized 12 compounds based on 8-PN and 6-prenylated naringenin (6-PN). Among the synthesized compounds, 8-PN and 6-PN showed high activity. Furthermore, this study shows that the position and steric structure of the prenyl group in the flavanone molecule are important. We also reported the intracellular kinetics of prenylated naringenins [38]. A prenyl or similar-structured group increased the affinity of the compound for the cell membrane. Although the multifunctionality of prenylated compounds was mentioned, 8-PN is noteworthy. We identified phytoestrogens, such as (a–c) flavonoids and (d) non-flavonoid compounds (Table 1 and Figure 4), involved in anti-influenza virus activity. Figure 4 showed the structure of phytoestrogen in WWMJ.

Phytoestrogen is a plant-derived compound, and functions similarly to endogenous estrogens *in vivo*. Daidzein, a phytoestrogen, has been reported to induce 5-hydroxyeicosatetraenoic acid, a compound involved in lipid metabolism, during viral infection [32]. The activation of MEK-ERK signaling may activate 5-lipoxygenase and thus induce 5-hydroxyeicosatetraenoic acid [33]. We hypothesize that 8-PN, which is a phytoestrogen [36], is also significantly involved in lipid oxidation. These results and reports suggest that the lipid metabolism of phytoestrogens is closely involved in antiviral responses. The elucidation of the influence of phytoestrogens on viral infection is necessary for studies on novel drug discovery. 

## 6. Conclusions and Perspectives

This review discussed the anti-influenza virus activity of wild watermelon. We previously reported the inhibitory effect of WWMJ on influenza virus growth. WWMJ inhibits virus entry and the late stage of virus replication in multiple reports. WWMJ inhibited viral entry and the late stages of viral replication and prolonged the survival of infected mice. The presence of various phytochemicals was analyzed through metabolomic analysis and triple quadrupole mass spectrometry. Its active ingredient is likely a low-molecular-weight compound. Our research group identified the phytochemicals, such as (a) acacetin, (b) naringenin, (c) prenylated flavonoids (8-PN), and (d) pinoresinol, were also involved in viral growth inhibition. 

The functionality of wild watermelon remained unclear until the 2000s; however, the mechanism of antioxidation has been studied and is gaining attention in Japan. The results obtained by Fujie et al. [48] suggested the possibility of a therapeutic approach utilizing wild watermelon for arteriosclerosis caused by lifestyle-related diseases. Our study suggests that unbred wild plants are a useful source for antiviral strategies. Ancient plants might have a high amount of active ingredients, but plants and foods are now being bred to suit our tastes. Other food functionality reports [68,69,70] also suggest that wild-type plants and foods have high functionality and may contain characteristic ingredients. To reap the benefits of plants and food, we should use varieties that are closer to their wild origins rather than plants bred to suit our tastes. Our study is the first to report on the novel functionality and anti-influenza virus effect of WWMJ (Figure 5). We emphasize that WWMJ is an excellent food material that can be utilized in multiple perspectives for the prevention and treatment of influenza. As WWMJ has a characteristic composition that contains phytochemicals such as phytoestrogen, in addition to its inhibitory effect on viral growth, we expect extensive research and development focused on wild watermelon in various fields in the future.

## Figures and Tables

**Figure 1 foods-12-03866-f001:**
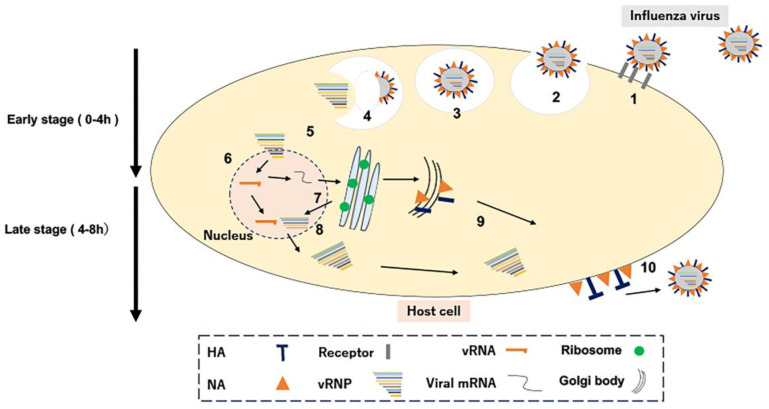
Viral growth steps in mammalian cells. Numbers 1–10 indicate the intracellular replication process of the influenza virus.

**Figure 2 foods-12-03866-f002:**
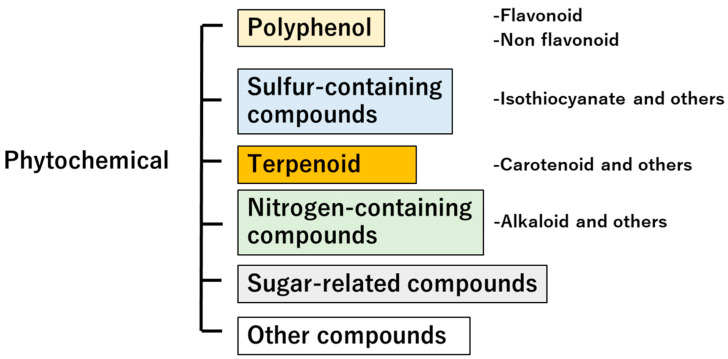
Plant-derived compound classification. The scheme shows the classification of compounds in the plant kingdom.

**Figure 3 foods-12-03866-f003:**
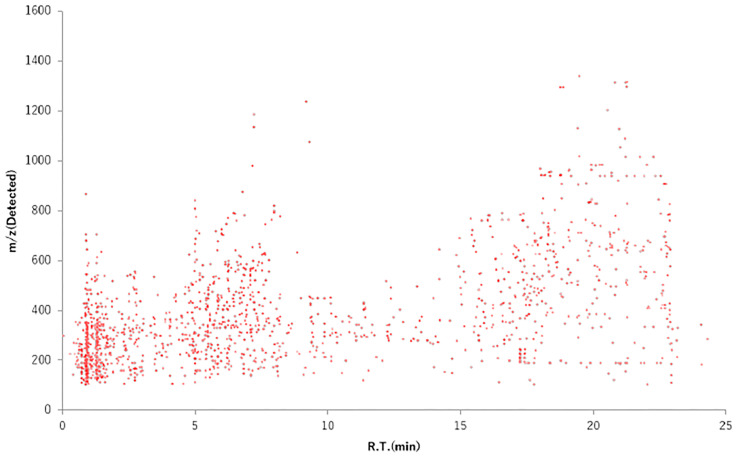
Low-molecular-weight compounds in WWMJ. Metabolome analysis was performed according to Hanada et al. [36]. WWMJ: wild watermelon juice.

**Figure 4 foods-12-03866-f004:**
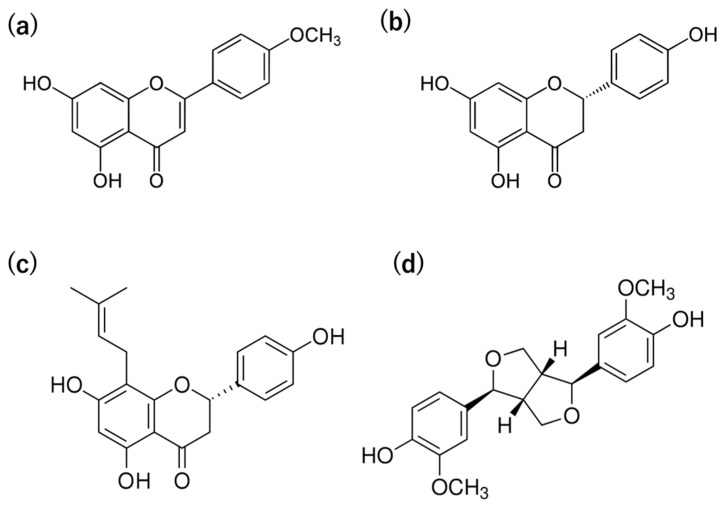
Structure of compounds, phytoestrogen in WWMJ. Metabolome and QQQ analysis system performed according to Hanada et al. [36]. (**a**) Acacetin. (**b**) Naringenin. (**c**) 8-prenylnaringenin. (**d**) Pinoresinol.

**Figure 5 foods-12-03866-f005:**
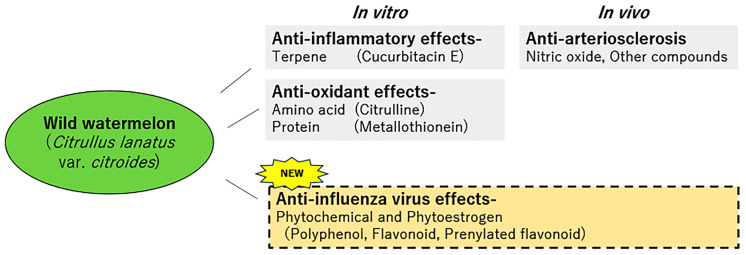
Novel food functionality of wild watermelon. The novelty of wild watermelon is summarized together with previous research reports.

**Table 1 foods-12-03866-t001:** Anti-viral effects of phytochemicals, phytoestrogen in WWMJ.

Classification	Compound Name and Type		IC_50_ Value (μM)
Flavonoid	Flavone	Acacetin (a)	33.8
Non-flavonoid	Flavanone Flavaprenin Lignan	Naringenin (b) 8-prenylnaringenin (c) Pinoresinol (d)	290.4 24.0 343.6

WWMJ was subjected to metabolome analysis and QQQ. The IC_50_ of flavonoids is reported as μM. These IC_50_ results were taken from Handa et al. and Morimoto et al. [36,37]. IC_50_: 50% inhibition concentration. WWMJ: wild watermelon juice. (**a**) Acacetin. (**b**) Naringenin. (**c**) 8-prenylnaringenin. (**d**) Pinoresinol.

## Data Availability

The data used to support the findings of this study can be made available by the corresponding author upon request.

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
