# Peer review of "Anti-Influenza Virus Activity of Citrullus lanatus var. citroides as a Functional Food: A Review"

_foods, 2023, doi:10.3390/foods12203866_

Round 1

Reviewer 1 Report

Comments and Suggestions for Authors

The article is a comprehensive review on the anti-influenza virus activity of Citrullus lanatus citroides. The work is very comprehensive and beautifully designed. The findings of the study shed light on future studies. In this context, working with minor corrections is acceptable.

Author Response

Reviewer1: The article is a comprehensive review on the anti-influenza virus activity of Citrullus lanatus citroides. The work is very comprehensive and beautifully designed. The findings of the study shed light on future studies. In this context, working with minor corrections is acceptable.

Response: Thank you for taking the time to critically evaluate our manuscript.

We sincerely appreciate your encouraging comments. We have added further details to the text based on the editor's comments and added more references, so we request that you review our revised manuscript.

Reviewer 2 Report

Comments and Suggestions for Authors

This review summarize the therapeutic properties of wild watermelon (Citrullus lanatus var. citroides) against influenza from a phytochemical view- point. Wild watermelon is a wild plant with significant potential as a therapeutic candidate in anti-viral strategies, when focused on its multiple anti-influenza functionality. Wild watermelon juice inhibits the viral growth, such as virus entry and replication. Hence, we highlight the possibility of utilizing wild watermelon for the prevention and treatment of influenza, with stronger antiviral activity. Phytochemicals and phytoestrogen (polyphenol, flavonoids, and prenylated compounds) in wild watermelon juice contribute to this activity and inhibit various stages of viral replication depending on the molecular structure. Wild plants and foods closely related to the original species contain many natural compounds such as phytochemicals and exhibit various viral growth inhibitory effects. These natural products provide useful information for future antiviral strategies. In my opinion, this review was well prepared and needs minor revision, as follow:

1.    The manuscript contains some spelling, grammatical and formatting mistakes (no spaces between many words) that should be revised carefully

2.    Structures of main cited compounds could be presented in an additional figure.

3.    It is not clear for me the importance of Figure 3. Please, this point must be well justified.

4.    The references should be carefully checked to be all in the same style.

5.    Authors need to include a strong Conclusion section to be able to make a statement about the "Future perspectives".

Comments on the Quality of English Language

The manuscript contains some spelling, grammatical and formatting mistakes (no spaces between many words) that should be revised carefully

Author Response

Reviewer2: This review summarize the therapeutic properties of wild watermelon (Citrullus lanatus var. citroides) against influenza from a phytochemical view- point. Wild watermelon is a wild plant with significant potential as a therapeutic candidate in anti-viral strategies, when focused on its multiple anti-influenza functionality. Wild watermelon juice inhibits the viral growth, such as virus entry and replication. Hence, we highlight the possibility of utilizing wild watermelon for the prevention and treatment of influenza, with stronger antiviral activity. Phytochemicals and phytoestrogen (polyphenol, flavonoids, and prenylated compounds) in wild watermelon juice contribute to this activity and inhibit various stages of viral replication depending on the molecular structure. Wild plants and foods closely related to the original species contain many natural compounds such as phytochemicals and exhibit various viral growth inhibitory effects. These natural products provide useful information for future antiviral strategies. In my opinion, this review was well prepared and needs minor revision, as follow:

  1. The manuscript contains some spelling, grammatical and formatting mistakes (no spaces between many words) that should be revised carefully

Response: Thank you for your suggestion. We have carefully checked the manuscript for grammatical and spelling errors.

  1. Structures of main cited compounds could be presented in an additional figure.

Response: Thank you for your comment. We detailed the structure of the compounds in Figure 4.

  1. It is not clear for me the importance of Figure 3. Please, this point must be well justified.

Response: Thank you for your suggestion. Figure 3 is necessary to understand the molecular weight distribution of compounds found in WWMJ. We have described the implications of Figure 3 in the revised text. We conducted a comprehensive analysis to examine the components contained in WWMJ.

  1. The references should be carefully checked to be all in the same style.

Response: Thank you for your suggestion. We have ensured that the references in the reference list have been uniformly formatted.

  1. Authors need to include a strong Conclusion section to be able to make a statement about the "Future perspectives".

Response: Based on your suggestion, we have added a statement on "Future perspectives" in the conclusion section. The revised section is indicated by red-colored text in the manuscript.

Reviewer 3 Report

Comments and Suggestions for Authors

1. Figure 2:  ETC.  ?????     this figure should be more detailed and ETC should be replaced with the other constituents

2. WWM???? why this abbreviation

3. WWMJ compounds     should be     WWMJ constituents

4. The chemical compounds should be drawn

5. If the authors can use the docking for suggesting of the biological activities of the compounds of this plant   6. Are there any reports described the any toxicity of this plant   7. I think it will be good if the authors can use ADMI software for study the possibility of using of the compounds of this plant as drug or no   8. The chemical structure of the components of this plant should be drawn and inserted in the manuscript within numbering    9. Are there any studies concerning the essential oils from this plant Comments on the Quality of English Language

Minor revisions needed

Author Response

Reviewer3: Minor revisions needed

  1. Figure 2: ETC. ????? this figure should be more detailed and ETC should be replaced with the other constituents.

Response: Based on your suggestion, the term "Etc" in Figure 2 has been corrected to “Other compounds".

  1. WWM???? Why this abbreviation

Response: Although WWM and WWMJ are mixed in the old manuscript, we have changed the abbreviations related to plants to only WWMJ in the revised manuscript. However, in Figure 5,

  1. WWMJ compounds should be WWMJ constituents.

Response: Thank you for your suggestion. As you say, WWMJ compounds have been replaced by WWMJ constituents in the revised manuscript.

  1. The chemical compounds should be drawn.

Response: Thank you for your comment. The chemical structures of the identified components are shown in Figure 4 of the revised manuscript.

  1. If the authors can use the docking for suggesting of the biological activities of the compounds of this plant.

Response: Thank you for your suggestion. Previous research reports include docking simulations of the identified flavonoids, which I have cited in the manuscript.

  1. Are there any reports described the any toxicity of this plant.

Response: Thank you for your comment. We have not encountered any papers regarding the toxicity of this plant. We also added a note to the revised manuscript regarding cytotoxicity to the manuscript.

  1. I think it will be good of the authors can use ADMI software for study the possibility of using of the compounds of this plant as drug or no.

Response: Thank you for your suggestion. For my next work, I would like to research the possible medicinal use of the compounds of this plant using ADMI software, as per your suggestion.

  1. The chemical structure of the compounds of this plant should be drawn and inserted in the manuscript within numbering.

Response: Thank you for your suggestion. We have included diagrams of the chemical structures of the major compounds of this plant in the revised manuscript.

  1. Are there any studies concerning the essential oils this plant.

Response: Thank you for this pertinent comment. We have not encountered any papers on the essential oils of this plant, although we found several papers related to fruit juice, as mentioned in the cited literature.
